# SIMPLE FAST CONVOLUTIONAL FEATURE LEARNING

## ABSTRACT

The quality of the features used in visual recognition is of fundamental importance for the overall system. For a long time, low-level hand-designed feature algorithms as SIFT and HOG have obtained the best results on image recognition. Visual features have recently been extracted from trained convolutional neural networks. Despite the high-quality results, one of the main drawbacks of this approach, when compared with hand-designed features, is the training time required during the learning process. In this paper, we propose a simple and fast way to train supervised convolutional models to feature extraction while still maintaining its high-quality. This methodology is evaluated on different datasets and compared with state-of-the-art approaches.

## 1 INTRODUCTION

The design of high-quality image features is essential to vision recognition related tasks. They are needed to provide high accuracy and scalability on processing large image data. Many approaches to building visual features have been proposed such as dictionary learning that aims to find a sparse representation of the data in the form of a linear combination of fundamental elements called atoms (Lazebnik et al., 2006). Scattering approaches provide mathematical frameworks to build geometric image priors (Oyallon & Mallat, 2015). Unsupervised bag of words methods identifies object categories using a corpus of unlabeled images (Sivic et al., 2005). Unsupervised deep learning techniques are also used to extract features by using neural networks based models with many layers and frequently trained using contrastive divergence algorithms (Le et al., 2011). All of them have been shown to improve the results of hand-crafted designed feature vectors such as SIFT (Lowe, 2004) or HOG (Dalal & Triggs, 2005) with promising results (Bo et al., 2010; Oyallon & Mallat, 2015).

Another recent but very successful alternative is to use supervised Convolutional Neural Networks (CNN) to extract high-quality image features (Razavian et al., 2014). These models take into consideration that images are symmetrical by a shift in position and therefore weight sharing and selective fields techniques are used to create filter banks that extract geometrically related features from the image dataset. The process is composed hierarchically over many layers to obtain higher level features after each layer. The network is typically trained using gradient backpropagation techniques. After the CNN training, the last layer (usually a fully connected layer) is removed to provide the learned features.

A drawback of this approach is the time needed to thoroughly train a CNN to obtain high accuracy results. In this paper, we propose the Simple Fast Convolutional (SFC) feature learning technique to significantly reduce the time required to learning supervised convolutional features without losing much of the representation performance presented by such solutions. To accelerate the training time, we consider few training epochs combined with fast learning decay rate.

To evaluate the proposed approach we combined SFC and alternative features methods with classical classifiers such as Support Vector Machines (SVM) (Vapnik, 1995) and Extreme Learning Machines (ELM) (Huang et al., 2006). The results show that SFC provides better performance than alternative approaches while significantly reduces the training time. We evaluated the alternative feature methods over the MNIST (Lecun & Cortes), CIFAR-10 and CIFAR-100 (Krizhevsky, 2009).

## 2   PROPOSAL

The training time required for a supervised convolutional approach to learning features from an image dataset is usually high. Indeed, it typically takes more than a hundred of training epochs to fully train a CNN to obtain high accuracy test results in classification tasks (Razavian et al., 2014). If the dropout technique is used, the process becomes even slower because much more epochs will probably be needed. However, this paper shows that as few as 10 epochs is generally enough to generate high-quality features.

In our proposal, the first innovation to learning feature fast is to design a step decay schedule that is a function of the total number of epochs predicted to train the model. For example, if the application needs to be trained in a time that is equivalent to only 10 training epochs, the SFC will provide a step decay schedule that is optimal to that amount of time. However, in the maximum training time allowed by the application permits to training 30 epochs, SFC will provide a step decay schedule that is optimal to provide the best possible accuracy performance training precisely 30 epochs. Taking in consideration the amount of time available to train to design an optimal step decay schedule is very import to develop a fast convolutional neural networks training procedure.

The second inspiration comes from the recent work of Shwartz-Ziv & Tishby (2017). The mentioned paper suggests that deep networks go through two learning phases. In the first one, the training happens very fast, while in the second one a slow progress represented by a fine-tuning takes place. In fact, Shwartz-Ziv & Tishby (2017) shows that at the beginning of training each layer is learning to preserve relevant input information. In this process, the mutual information between the representation of each layer and the relation input/output is enhanced in some cases almost to linear. The next phase, in consequence, stabilizes the mutual information between each layer and the output allowing each layer to adapt its weights to prioritize the information that is important for the output mapping.

In this sense, considering previews mentioned results, we assume that preserving for a large percentage of epochs this first phase of the network training, which has significant gradient means and small variance, is very important to generate high-quality features fast. Regarding our proposal, it suggests that an initial high learning rate has to be kept for a high percentage of training epochs.

Regarding the second phase of the training of deep neural networks (Shwartz-Ziv & Tishby, 2017), we understand that concerning step decay scheduling the opposite should happen. For this stage, we consider exponential and fast decay of the learning rate to produce high-quality fine-tuning of the models. In this sense, after keep the initial high learning rate for a considerable percentage of epochs, we start exponential decays faster and faster.

### 2.1   STEP DECAY SCHEDULE

Taking into consideration the above explanations, we define SFC as follows. Given a number of epochs to train, we set up the first learning rate decay after 60% of the predicted training epochs. After, a new learning rate decay should happen at 80% of the total scheduled of epochs. Finally, the last learning rate decay is set to be performed at the 90% of the chosen number of epochs. For example, if the number of epochs selected is 30, the learning rate decay should happen at epochs 18, 24 and 27. We define the learning rate decay to be 0.2. These values were obtained and validated experimentally considering different datasets, models, architectures, and parameters.

### 2.2   FEATURE EXTRACTION

After we train the convolutional network and therefore learning the features, the last fully connected layer is dropped, exposing the nodes of the previous layer. Given a target image, we apply the example to the input layer of the learned model, and the network output is the high-level Simple Fast Convolutional (SFC) feature representation. After extracting the SFC features some classical classifier can be used to construct the decision surface as proposed by works such as LeCun et al. (1998) and Huang & LeCun (2006).

## 3 EXPERIMENTS

The MNIST is a dataset of 10 handwritten digits classes. In our experiments, we used 60 thousand images for training and 10 thousand for the test. Each example consists of a grayscale 28x28 pixels image. The CIFAR-10 dataset consists of 10 classes, each containing 6000 32x32 color images, totalizing 60000. There are 5000 training images and 1000 test images for each class. The CIFAR-100 dataset has 100 classes containing 600 images each. For training, there are 500 images per class while for the test there are 100 images per class.

For MNIST dataset, we are using as baseline model the LeNet5 (LeCun et al., 1998). Our modifications to the original model were changing the activation function to ReLU and adding the batch normalization to the convolutional layers. Despite being a tiny and almost twenty years old model, the proposed LeNet5 is capable of presenting a high performance concerning training time and test accuracy with only 10 training epochs.

The CIFAR-10 and CIFAR-100 dataset were trained with an adapted Visual Geometry Group (Simonyan & Zisserman, 2014) type model. We designed the system with nineteen layers and batch normalization but without dropout (VGG19). In both cases, to extract for each image a 256 linear feature vector, the last layer before the full connected classifier was changed to present 256 nodes.

The initial learning rate was 0.1. We used stochastic gradient descent with Nesterov acceleration technique and moment of 0.9 as the optimization method. The chosen weight decay was 0.0005. The mini-batches were set to have the size of 128 images each. Since one of our objectives is to produce a fast solution with smaller training times while keeping a high accuracy, we did our experiments avoiding any data augmentation (distortion, scaling or rotation).

We used SVM and ELM as base classifiers to evaluate the quality of features compared in this study. For SVM we used the default values ($C = 1$ and the gamma value is the inverse of the number of features) and regular Gaussian Kernels. For ELM, we used 1024 nodes in the hidden layer. In the following figures and tables, we used the label SVM to refer experiments that consist of using the SVM classifier with Gaussian kernel directly into a flat array of raw image pixels.

Similarly, we used the label ELM to refer experiments that use an ELM classifier with 1024 hidden nodes applied directly to the flat array of raw pixel values. The $SVM_{SFC10}$ refers to experiments with the SVM classifier using SFC features trained during 10 epochs and similarly $ELM_{SFC10}$ means that the experiments using ELM classifiers, instead of being trained in raw pixels, use the SFC features obtained after 10 CNN training epochs.

If no classifier is mentioned at all, as in the expression SFC10, then the original CNN classifier (the last fully connected layer) was kept as an example of a classifier in this particular case. Therefore, SFC30 represent the performance of the CNN itself trained for 30 epochs using the SFC training schedule. In such situations, we compared multiple variations of Fast Convolution (SFC10, SFC30, and SFC100) to the classical training schedule approach, which uses numerous training epochs with a fixed learning rate.

Therefore, in the first classical training schedule ($CNN_{0.1}$), we trained the models during 100 epochs with a constant learning rate of 0.1. In the second regular training schedule ($CNN_{0.01}$), the models were trained throughout 100 epochs with a learning rate of 0.01.

The experiments were performed using a Linux Ubuntu machine with Intel(R) Core(TM) i7-4790K CPU @ 4.00GHz, 16GB RAM memory, 1T hard drive, and a NVIDIA GeForce GTX 980 Ti.

## 3.1 MNIST

In Table 1, we compared the test accuracy and training time of $SVM_{SFC10}$ with H-ELM (Tang et al., 2016), a speedy approach to expand ELM to multilayer networks. SFC10 not only produces better accuracy but also reduces the training time considerably. In Table 2, the experiments show that SFC10 provides test accuracy similar to $CNN_{0.1}$ and $CNN_{0.01}$, despite being about 10 times faster.

Table 1: MNIST performance comparison

| Method | Test Accuracy(%) | Training Time(s) |
|---|---|---|
| H-ELM (Tang et al., 2016) | 99.13 | 281.37 |
| $SVM_{SFC10}$ | **99.39** | **27.98** |

Table 2: MNIST Simple Fast Convolution performance

| Method | Test Accuracy(%) | Training Time(s) |
|---|---|---|
| SFC10 | 99.37±0.05 | 15.28±0.75 |
| SFC30 | 99.38±0.05 | 46.88±0.97 |
| SFC100 | 99.46±0.02 | 154.40±0.14 |
| $CNN_{0.1}$ | 99.24±0.04 | 164.49±1.86 |
| $CNN_{0.01}$ | 99.22±0.07 | 154.57±0.34 |

## 3.2 CIFAR-10

The Fig. 1 presents the experiments on CIFAR-10. The SVM, ELM, and even H-ELM present low accuracy when trained on raw features. It happens because CIFAR-10, unlikely MNIST, is a non-sparse dataset and since the importance of using high-quality features is high in this case. The use of the SFC features significantly reduces the training time. Moreover, the SVM and ELM classifiers improved their performance on the test set in about 30% and 50%, respectively.

The comparative study is shown in the Table 3, where it can be seen that SFC features achieve the best performance when compared with alternative approaches. Naturally, the test accuracy can go even higher if the training time is not an issue and more epochs could be used.

The Table 4 shows how fast the SFC can learn high-quality representations if the final objective is not the absolute maximum possible test accuracy. The mentioned table compare the test accuracy of $SVM_{SFC10}$, $ELM_{SFC10}$, and the original VGG19 model. All results presented are a mean of 10 runs. The VGG19 model was trained with the aim to achieve the best possible accuracy. In this sense, it was trained with 100 epochs, and learning rate decays at epochs 60, 80 and 90, which is equivalent to the SFC100 training schedule. The results show that about 95% of the final accuracy can be achieved with approximately 10% of the training time.

In table 5, the experiments showed that SFC10 provided better test accuracy than both $CNN_{0.1}$ and $CNN_{0.01}$, despite being about 10 times faster. If training time is not an issue, even better test accuracy can be obtained with slower SFC variants.

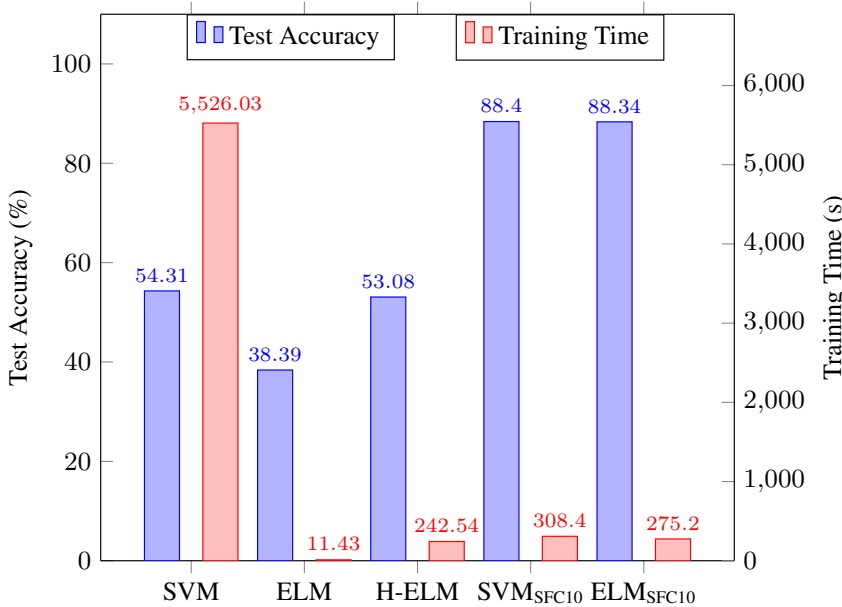

Figure 1: Mean test accuracy and training time of 10 runs on CIFAR-10 dataset.

Table 3: CIFAR-10 performance comparison

| Method | Type | Test Accuracy(%) |
|---|---|---|
| SIFT (Bo et al., 2010) | Prior Knowl. | 65.6 |
| LIFT (Sohn & Lee, 2012) | Unsup. Deep | 82.2 |
| RotoTrans. Scat. (Oyallon & Mallat, 2015) | Prior Knowl. | 82.3 |
| NOMP (Lin & EDU, 2014) | Unsup. Dict. | 82.9 |
| RFL (Yangqing Jia et al., 2012) | Unsup. Dict. | 83.1 |
| SVM$_{SFC10}$ | Supervised | **88.40±0.24** |

Table 4: CIFAR-10 fully trained VGG19 and Simple Feature Convolutional

| Model | Accuracy(%) [VGG19(%)] | Time(s) [VGG19(%)] |
|---|---|---|
| SVM$_{SFC10}$ | 88.40 [**95.88**] | 308.40 [**12.06**] |
| ELM$_{SFC10}$ | 88.34 [**95.82**] | 275.20 [**10.76**] |
| VGG19 | 92.19 [100.00] | 2556.29 [100.00] |

Table 5: CIFAR-10 Simple Fast Convolution performance

| Method | Test Accuracy(%) | Training Time(s) |
|---|---|---|
| SFC10 | 88.44±0.27 | 266.18±0.86 |
| SFC30 | 91.17±0.14 | 805.83±0.88 |
| SFC100 | 92.19±0.19 | 2556.29±3.74 |
| CNN$_{0.1}$ | 85.12±0.37 | 2611.74±7.52 |
| CNN$_{0.01}$ | 87.57±0.10 | 2571.10±6.58 |

### 3.3 CIFAR-100 EXPERIMENTS

The Fig. 2 shows that training on raw data produces even worst results on this dataset and that SFC, in fact, improves the performance of the test accuracy by providing high-quality features to the classifiers. Once again, the results presented by the two used classifiers are mostly the same, with all the differential of the solution being the feature representation used.

The comparative study on dataset CIFAR-100 is shown in the Table 6. Once again, the SFC feature learning approach produces the best results. Our experiments showed that SFC30 provided better test accuracy than both $CNN_{0.1}$ and $CNN_{0.01}$, despite being about three times faster (Table 7).

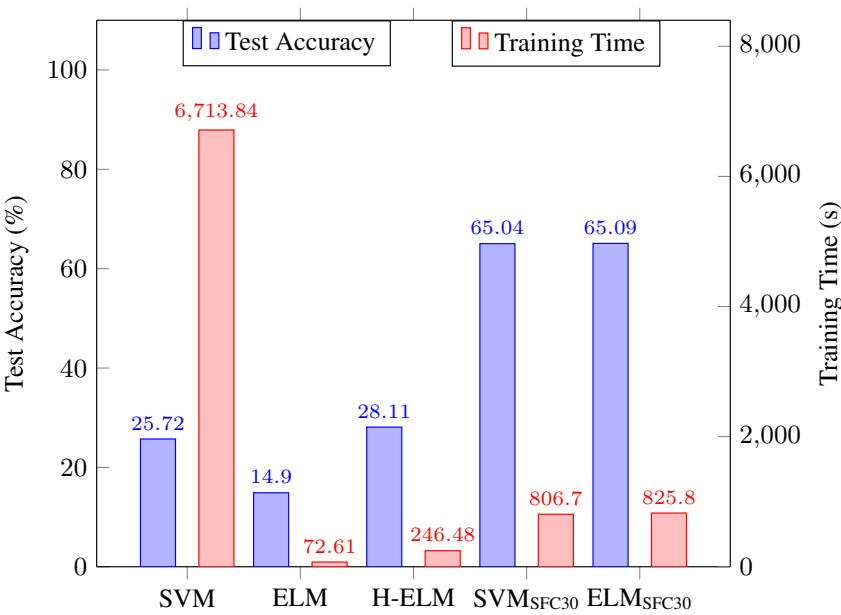

Figure 2: Mean test accuracy and training time of 10 runs on CIFAR-100 dataset.

Table 6: CIFAR-100 performance comparison

| Method | Type | Test Accuracy(%) |
|---|---|---|
| RFL (Yangqing Jia et al., 2012) | Unsup. Dict. | 54.2 |
| RotoTrans. Scat. (Oyallon & Mallat, 2015) | Prior Knowl. | 56.8 |
| NOMP (Lin & EDU, 2014) | Unsup. Dict. | 60.8 |
| $SVM_{SFC30}$ | Supervised | **65.03±0.31** |

Table 7: CIFAR-100 Simple Fast Convolution performance

| Method | Test Accuracy(%) | Training Time(s) |
|---|---|---|
| SFC10 | 52.96±1.01 | 256.87±1.37 |
| SFC30 | 65.40±0.27 | 798.42±2.01 |
| SFC100 | 69.31±0.38 | 2554.85±2.99 |
| $CNN_{0.1}$ | 54.26±0.25 | 2545.86±3.67 |
| $CNN_{0.01}$ | 59.27±0.41 | 2559.71±6.50 |

The results presented in the Tables 2, 5 and 7 essentially show that, when compared to classical approach of numerous epochs of constant learning rate ($CNN_{0.1}$ and $CNN_{0.01}$), the SFC variants provide better test accuracy despite being many times faster. Therefore, a fast changing in the learning rate is indeed a viable alternative to speed up deep neural networks training time.

## 4    CONCLUSION

In this paper, we showed that convolutional feature learning can be performed in a fast way. Moreover, despite being very fast, it is still capable of generating representations that present better performance than other approaches. The proposed method is also flexible enough since a compromise can be obtained between the speed of the training and the final solution test accuracy. Naturally, the difference in test accuracy presented in this paper could be even greater if more training time is allowed to be used.

We emphasize that transfer learning techniques can be used to extend the application of the proposed method. Finally, we show that despite efforts to the contrary, supervised convolutional method still provides state-of-the-art results for image feature generation. Moreover, the experiments showed that a quick change in the learning rate decay is a valid method to speed up the training of deep neural networks significantly.

ACKNOWLEDGMENT.

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
