# OpenReview forum: "Simple Fast Convolutional Feature Learning"
_ICLR.cc/2018/Conference — Reject_

### Official Review · AnonReviewer1 · 2017-11-25
**Lack of significant results**

**Rating:** 3
**Confidence:** 4

**Review:**

This paper deals with early stopping but the contributions are limited. This work would fit better a workshop as a preliminary result, furthermore it is too short. Following a short review section per section.

Intro: The name SFC is misleading as the method consists in stopping early the training with an optimized learning schedule scheme. Furthermore, the work is not compared to the appropriate baselines.

Proposal: The first motivation is not clear. The training time of the feature extractor has never been a problem for transfer learning tasks for example: once it is trained, you can reuse the architecture in a wide range of tasks. Besides, the training time of a CNN on CIFAR10 or even ImageNet is now quite small(for reasonable architectures), which allows fast benchmarking.
The second motivation, w.r.t. IB seems interesting but this should be empirically motivated(e.g. figures) in the subsection 2.1, and this is not done.

The section 3 is quite long and could be compressed to improve the relevance of this experimental section. All the accuracies(unsup dict, unsup, etc) on CIFAR10/CIFAR100 are reported from the paper (Oyallon & Mallat, 2015), ignoring 2-3 years of research that leads to new numerical results. Furthermore, this supervised technique is only compared to unsupervised or predefined methods, which is is not fair and the training time of the Scattering Transform is not reported, for example.

Finally, extracting features is mainly useful on ImageNet (for realistic images) and this is not reported here.

I believe re-thinking new learning rate schedules is interesting, however I recommend the rejection of this paper.

---

### Official Review · AnonReviewer2 · 2017-11-26
**No contribution**

**Rating:** 3
**Confidence:** 4

**Review:**

This paper proposes a fast way to learn convolutional features that later can be used with any classifier. The acceleration of the training comes from a reduced number of training epocs and a specific schedule decay of the learning rate.
In the evaluation the features are used with support vector machines (SVN) and extreme learning machines on MNIST and CIFAR10/100 datasets.

Pros:
The paper compares different classifiers on three datasets.

Cons:
- Considering an adaptive schedule of the learning decay is common practice in modern machine learning. Showing that by varying the learning rate the authors can reduce the number of training epocs and still obtain good performance is not a contribution and it is actually implemented in most of the recent deep learning libraries, like Keras or Pytorch.
- It is not clear why, once a CNN has been trained, one should want to change the last layer and use a SVN or other classifiers.
- There are many spelling errors
- Comparing CNN based methods with hand-crafted features as in Fig. 1 and Tab.3 is not interesting anymore. It is well known that CNN features are much better if enough data is available.

---

### Official Review · AnonReviewer3 · 2017-11-30
**A simple combination of known approaches?**

**Rating:** 2
**Confidence:** 4

**Review:**

I am not sure how to interpret this paper. The paper seems to be very thin technically, unless I missed some important details. Two proposals in the paper are:

(1) Using a learning rate decay scheme that is fixed relative to the number of epochs used in training, and
(2) Extract the penultimate layer output as features to train a conventional classifier such as SVM.

I don't understand why (1) differs from other approaches, in the sense that one cannot simply reduce the number of epochs without hurting performance. And for (2), it is a relatively standard approach in utilizing CNN features. Essentially, if I understand correctly, this paper is proposing to prematurely stop training an use the intermediate feature to train a conventional classifier (which is not that away from the softmax classifier that CNNs usually use). I fail to see how this would lead to superior performance compared to conventional CNNs.

---

> ### Author Response · Authors · 2017-12-10
> **Comment answers**
>
> Dear Reviewer,
>
> In the following lines, we try to clarify your doubts.
>
> You wrote: "I don't understand why (1) differs from other approaches, in the sense that one cannot simply reduce the number of epochs without hurting performance."
>
> Transfer learning and domain adaptation are essential in machine learning. Indeed, in some situation, we have a small image dataset which is not able to be used to train a Convolutional Neural Network (CNN) thoroughly. In such cases, we can either use a hand-designed feature or extract features from a CNN pretrained in a large dataset.
>
> Despite being a standard approach today, extracting features from a CNN to performing transfer learning or domain adaptation has a significant drawback when compared with using hand-designed features: the training time required. Indeed, hand-designed features do not need to be trained and since they are immediately available. Hence, despite usually provide higher quality features (when a large dataset is available), extracting features from a CNN takes much more time than using directly available engineered features.
>
> Therefore, this work aims to show that we can significantly mitigate this drawback by showing that it is possible to dramatically reduce the training time required to pretrain a CNN without significantly affect the quality of the generated features.
>
> Hence, we propose a method that is very efficient in considerably reducing the training time need to extract features from a CNN with minor impact on the quality of the generated features.
>
> Trading a significant training time reduction by a small decrease in features quality reduces the above mention drawback of using CNN feature extraction rather than hand-designed ones.
>
> The proposed approach, despite simple, innovates in showing that a learning schedule that is aware of the available training can maximize its use with minor performance hurting. In other words, we propose a simple way to produces high-quality features given a time constraint requirement.
>
> You wrote: "And for (2), it is a relatively standard approach in utilizing CNN features. Essentially, if I understand correctly, this paper is proposing to prematurely stop training an use the intermediate feature to train a conventional classifier (which is not that away from the softmax classifier that CNNs usually use). I fail to see how this would lead to superior performance compared to conventional CNNs."
>
> I believe the previous explanation clarifies this point. The objective is not to produce a better performance than it would be possible if more time were available. The aim is to show that the proposed method can provide almost the best possible feature quality in a small fraction of the time it would require to thoroughly train the CNN in order to get the utterly best possible feature quality.

---

### Public Comment · (anonymous) · 2017-11-28
**Clarification questions**

We’re trying to reproduce your work and we found certain ambiguities which we’d like to have clarified.

You have mentioned that for the MNIST dataset, you used the 60k – 10k split, however, it is not clear whether you have used the default split or you created your own (e.g. merged all data, shuffled them and split them).

We are also not sure how you used SVM for classifying samples that belong to 10 different classes – did you use one-vs-all approach or multiclass classification?

Regarding the underlying structure of SFC, we assume that LeNet5 with ReLu activation units is used since it is the only one mentioned in the paper that is not used for the baseline classifiers – is this assumption correct?

Could you also provide us with some additional information regarding which (if any) libraries or frameworks have been used for constructing the learning algorithms? Alternatively, if you happen to have the code publicly available, it would be helpful if you could direct us to it.

Thank you for the answers in advance.

---

> ### Author Response · Authors · 2017-11-29
> **Comment answers**
>
> First of all, thank you for your interest in our work.
>
> Regarding your questions, we have used the default split for the MNIST dataset. Referring to SVM, we have used the multiclass variant.
>
> Despite being clear in the paper, we confirm that we have used ReLU in our adapted LeNet5 model. The libraries and frameworks which we have used were Numpy, Scikit-learn, Pandas, and PyTorch.
>
> Best regards.

---

### Public Comment · (anonymous) · 2017-12-09
**Reproducibility of MNIST and CIFAR-10 results**

The paper under review investigates the extraction of features from images for recognition and classification purposes. The authors of the paper propose a method to simplify and increase the speed of feature extraction using convolutional models while pointing out that the drawback to this is the time required to train a CNN. Therefore, they also propose a scheduling technique in order to accelerate the training process, while maintaining the performance level.

In order to reproduce their results, we had to make numerous assumptions since the paper lacks certain information about the setup of the SFC and more. Although not entirely clear which layout is used where, we assumed that LeNet-5 was used to represent SFC for MNIST dataset and VGG19 for CIFAR datasets. For the datasets, we assumed no preprocessing was done and that the default train / test splits were used.

We were able to reproduce most of the results for the two datasets that we tried to verify (i.e. MNIST and CIFAR-10), however, we could not confirm some outcomes that were obtained by the authors. Most notably, it seems that the SFC scheduling does not work very well with LeNet-5 CNN since we experienced a sudden drop in accuracy at around epoch 50 and therefore the final accuracy for SFC100 and CNN0.1 was only 10%. We ran this test 10 times and this behavior was exhibited in each one of the rounds. Regarding the training times claimed by the authors, our experiments took roughly twice as much time for the MNIST dataset, which could be explained by less computational power at hand, however, the CIFAR-10 training times we saw were almost 5 times slower. Since we did not have access to the original code that the authors have used and some hyperparameters were not mentioned in the paper (such as number of iterations per epoch for VGG19), this could serve as an explanation for the prolonged training times and irreproducibility of the MNIST experiments.

Overall the paper does not provide us with enough information on the setup and architectures to reliably reproduce the results observed by the authors. For more details on the code that we have used, visit https://github.com/lgatting/AML2017-Assignment-4.

---

> ### Author Response · Authors · 2017-12-10
> **Comment answers**
>
> Dear anonymous,
>
> Please, be more specific instead of saying that "we had to make numerous assumptions since the paper lacks certain information about the setup of the SFC and more".
>
> For example, you said that "Although not entirely clear which layout is used where, we assumed that LeNet-5 was used to represent SFC for MNIST dataset and VGG19 for CIFAR datasets". However, the paper is very clear in saying that "For MNIST dataset, we are using as baseline model the LeNet5 (LeCun et al., 1998)". Moreover: "The CIFAR-10 and CIFAR-100 dataset were trained with an adapted Visual Geometry Group (Simonyan & Zisserman, 2014) type model".
>
> Regarding preprocessing, for either MNIST and CIFAR10/100, standard mean-std was used. Moreover, regarding CIFAR10/100, we also performed a random horizontal flip with probability 0.5. We will update to paper regarding this information.
>
> We saw in code and it appears that you are not using batch normalization for LeNet5, but we used it in our experiments. In the paper, we wrote: "For MNIST dataset, we are using as baseline model the LeNet5 (LeCun et al., 1998). Our modifications to the original model were changing the activation function to ReLU and adding the batch
> normalization to the convolutional layers."
>
> Regarding VGG19, you appear to use dropout (p=0.5), despite we write in the paper that "We designed the system with nineteen layers and batch normalization but without dropout (VGG19)". Besides, you are using the original VGG19 variant used for ImageNet which has three fully connected layers instead of the VGG19 variation regularly used with CIFAR10/100 which has just one fully connected layer. Please, for example, visit https://github.com/kuangliu/pytorch-cifar/blob/master/models/vgg.py
>
> Finally, you need to pay attention to this line of the paper: " In both cases, to extract for each image a 256 linear feature vector, the last layer before the full connected classifier was changed to present 256 nodes" as your code does not appear to follow this instruction.
>
> Regarding the comment "some hyperparameters were not mentioned in the paper (such as number of iterations per epoch for VGG19)", we would like to clarify that the number of interactions per epoch is deterministically determined by the size of the training set and the batch size. Both pieces of information are in the paper for either MNIST and CIFAR10/100. Therefore, we should correct your code and use 60000/128=469 for MNIST and 50000/128=391 for CIFAR10/100.
>
> We are using an NVidia 980i TI which has 6GB of RAM. If we are using a card with less memory or cores, this is probably the reason why your experiments are slower than ours.
>
> Again, we asked you to be more specific instead of writing "Overall the paper does not provide us with enough information on the setup and architectures to reliably reproduce the results observed by the authors".

---

### Decision · Program_Chairs · 2018-01-29
**ICLR 2018 Conference Acceptance Decision**

**Decision:**

Reject

**Comment:**

The paper addresses the training time of CNNs, in the common setting where a CNN is trained on one domain and then used to extract features for another domain.  The paper proposes to speed up the CNN training step via a particular proposed training schedule with a reduced number of epochs.  Training time of the pre-trained CNN is not a huge concern, since this is only done once, but optimizing training schedules is a valid and interesting topic of study.   However, the approach here does not seem novel; it is typical to adjust training schedules according to the desired tradeoff between training time and performance.  The experimental validation is also thin, and the writing needs improvement.